# Effects of the Antioxidant Quercetin in an Experimental Model of Ulcerative Colitis in Mice

**DOI:** 10.3390/medicina59010087

**Published:** 2022-12-31

**Authors:** George Kottakis, Katerina Kambouri, Alexandra Giatromanolaki, Georgia Valsami, Nikolaos Kostomitsopoulos, Alexandra Tsaroucha, Michael Pitiakoudis

**Affiliations:** 1Department of Experimental Surgery, Democritus University of Thrace, 68100 Alexandroupolis, Greece; 2Department of Pediatric Surgery, Democritus University of Thrace, 68100 Alexandroupolis, Greece; 3Department of Pathology, Democritus University of Thrace, 68100 Alexandroupolis, Greece; 4Department of Pharmacy, School of Health Sciences, National and Kapodistrian University of Athens, 15784 Athens, Greece; 5Biomedical Research Foundation (BRFAA) of the Academy of Athens, Experimental Surgery and Translational Research, 11527 Athens, Greece; 62nd Department of General Surgery, University of Thrace, 68100 Alexandroupolis, Greece

**Keywords:** quercetin, antioxidant, ulcerative colitis, model, mice

## Abstract

*Background and Objectives*: Quercetin, a member of the flavanol family found in many fruits, vegetables, leaves and grains has been found to have a wide range of biological effects on human physiology. The aim of this study was to investigate the effects of quercetin, when administered orally in the form of the water-soluble inclusion complex with hydroxypropyl-b-cyclodextrin (Que-HP-β-CD), in an experimental model of ulcerative colitis in mice. *Materials and Methods*: Animals received either Dextran Sodium Sulphate (DSS), to induce colitis, + Que-HP-β-CD (Group A), DSS alone (Group B) or no intervention (control, Group C) for 7 days. All animals were weighed daily, and evaluation of colitis was performed using the Disease Activity Index (DAI). On day 7 a blood sample was taken from all animals, they were then euthanised, the large intestine was measured, and histological and immunochemical analyses were performed. *Results*: The DAI demonstrated an increase over time for the groups receiving DSS (Groups A and B) compared with the control group (Group C), with a significant degree of protection being observed in the group that also received quercetin (Group A): The DAI over time slope for Group B was higher than that for Group A by 0.26 points/day (95% Cl 0.20–0.33, *p* < 0.01). Weight calculations and immunohistochemistry results validated the DAI findings. *Conclusions*: In conclusion, the administration of quercetin in an ulcerative colitis model in mice presents a therapeutic/prophylactic potential that warrants further investigation.

## 1. Introduction

Ulcerative colitis (UC) is a chronic inflammatory disorder of the large intestine. Although its aetiology remains unknown, the consensus is that it is a multifactorial disease linked to genetic predisposition, environmental factors, irregularities in intestinal microbiota and inappropriate immune system response [1]. Clinical studies have also shown increased levels of reactive oxygen molecules (ROMs) and a decrease in antioxidants in patients [2]. This oxidative stress, apart from its end-result of tissue inflammation, is thought to have a possible aetiological role in the pathogenesis of UC [3,4]. The main symptom of UC is bloody diarrhea with mucus, with or without abdominal pain. It is accompanied by loss of energy, appetite and weight, and is characterized by exacerbations and remissions of these symptoms. Mucosal damage starts at the rectum and extends proximally in a continuous manner to varying lengths of the large intestine, with a distinct demarcation line evident between inflamed and healthy mucosa. If the disease is limited to the rectum it is classified as proctitis, if it extends to the splenic curve, it is classified as left sided colitis and if it covers the whole length of the large intestine as pancolitis [5]. Proctitis is characterized by frequent bloody diarrhea with mucus, accompanied by pain and tenesmus. In acute left sided colitis, or pancolitis, patients may have up to 20 bloody bowel movements per day, which are accompanied by incontinence during the night. Such episodes often require hospitalization. The extra-intestinal manifestations (EIMs) of UC are also of significance, as the disease can affect the joints, causing arthritis and ankylosing spondylitis, the skin, causing pyoderma gangrenosum and nodular erythema, the eyes and the hepato-biliary system, with the most serious EIM being primary sclerosing cholangitis. In active colitis there is an increase in White Blood Cells (WBC), C-Reactive Protein (CRP) and Erythrocyte Sedimentation Rate (ESR), reflecting the inflammatory condition. CRP levels above 23 mg/L at diagnosis indicate severe disease and are predictive of the eventual need of surgery [6]. In children, a combination of the WBC count, ESR, and either platelets (PLT) or albumin has been reported to be the best predictive subset of standard laboratory tests to identify severe from non-severe clinical or mucosal disease at diagnosis [7]. Despite recent progressive leaps in the development of therapeutic protocols for the management of UC, it still exerts a significant strain on the worldwide medical community [8] as patients often need chronic treatment to remain in remission and in those cases that fail to respond to available medication can lead to colectomy.

Current treatment of UC depends on disease severity, which is measured by clinical scoring and endoscopic examination [9,10]. For mild to moderate disease, mesalamine is considered as first-line treatment as it can promote remission in a large percentage of patients and can be administered orally, by suppository or as an enema [11]. For unresponsive patients, the next step is corticosteroid [12] or immunosuppressive therapy [13], or a combination of the two. However, these are not appropriate for maintenance therapy as they are accompanied by a range of adverse effects, such as corticosteroid dependence/resistance [12], osteoporosis and adrenal insufficiency attributed to chronic corticosteroid therapy [14], and nausea, vomiting, leukopenia, pancreatitis and hepatitis [15] attributed to immunosuppressive therapy.

For moderate to severe disease, the development of monoclonal antibody therapies has recently changed the playing field as these treatments can target inflammatory cytokines (infliximab) [16] or receptor proteins on the surface of lymphocytes (vedolizumab) [17,18], thus regulating the pathological immune response. However, some patients fail to respond or to remain in remission with these advanced therapies [19], creating the need for chronic treatment with corticosteroids or surgical management, and the need to evaluate alternative options.

Currently, there is growing interest in the scientific evaluation of natural products found in traditional medicine. These cheap, easily accessible agents, with minimal adverse effects, can be utilized as alternative treatment options and as prophylactic agents for a range of conditions [20]. Natural antioxidants, such as ginger extract and crocus sativus stigmas, have been found to be beneficial as anti-hyperglycaemics [21], anticonvulsants [22] and antidepressants [23], amongst various other uses. Several studies have examined the use of naturally derived antioxidants on experimental models of diseases of the bowel, such as necrotising enterocolitis [24,25], colitis [26,27] and colonic cancer [28]. The promising results of these studies has stimulated the search for other relevant active ingredients to further our therapeutic arsenal and our knowledge of the disease. One such promising option is quercetin (Figure 1), a member of the flavanol family present in many fruits, vegetables, leaves and grains, making it readily available in the daily dietary intake of humans. It has been extensively studied and found to have a wide range of biological effects on human physiology, including antioxidant [29], anti-inflammatory [30], antiallergic [31], antiviral [32] and anticancer [33] properties. The potential utilization and evaluation of its effectiveness as an antioxidant, has been performed through free-radical scavenging, and its anti-inflammatory attributes, through inhibition of tumor necrosis factor α (TNF-α) [34], cyclooxygenase (COX) and lipoxygenase (LOX) production [35,36]. Such studies can hopefully help investigate the benefits of quercetin consumption and its application as a possible therapeutic option in various diseases. 

However, the main problem with the pharmaceutical application of Que is its low aqueous solubility, namely about 1 mg/mL in water, 5.5 mg/mL in simulated gastric fluid and 28.9 mg/mL in simulated intestinal fluid [37], that leads to poor oral bioavailability. Several studies have reported the enhanced Que solubility upon complexation with cyclodextrins and, in particular, with hydroxypropylβ-cyclodextrin (HP-b-CD) [38,39,40]. Recently, Kelici et al. [41] synthesized a Que-HP-β-CD complex that enabled amplification of its solubility while retaining its bioactivity in T24 human bladder cancer cell line. The same research group [42] also evaluated Que-HP-β-CD complex through solubility and dissolution experiments, Cyclic Voltammetry, UV–Vis spectroscopy, HPLC-ESI-MS/MS and HPLC-DAD, fluorescence spectroscopy, NMR Spectroscopy and theoretical calculations (density functional theory, DFT), while biological evaluation of the protection offered against H_2_O_2_-induced DNA damage was also performed. The authors found that encapsulation of quercetin inside the HP-β-CD cavity enhanced its solubility and retained its oxidation profile, while iron can operate as a chemical stimulus to release quercetin from its host cavity.

The aim of this study was to investigate the potential protective effects of quercetin on UC, when administered orally in the form of the recently developed water-soluble lyophilized inclusion complex with hydroxypropyl-β-cyclodextrin (Que-HP-β-CD) [41,42] on an experimental model of UC in mice, for possible future therapeutic use in humans, using the Disease Activity Index (DAI), as well as tissue histology and immunohistochemistry for integrin a4β7, Mucosal vascular addressing cell adhesion molecule-1(MAdCAM) and CD11b tissue levels, as proof of concept.

## 2. Materials and Methods

### 2.1. Quercetin

The Que-HP-β-CD lyophilized product was prepared as previously described [41,42] in molar ratio of 1:2 using the neutralization method [43] Briefly, 4.8 g of HP-β-CD was weighed accurately, transferred into a 600 mL beaker, and suspended with 500 mL of water, until the complete solubilization of the CD. 500 mg of Que was transferred to the beaker under continuous stirring and light protection of the photosensitive Que, and small amounts of ammonium hydroxide were added until complete dissolution of Que. The pH was continuously monitored and adjusted to approximately 9−9.5. The obtained solution was divided into round trays for lyophilization and frozen at −73 °C to further be freeze-dried using a Biobased Vacuum Freeze-Dryer, BK-FD10T, Biobase biodustry (Shandong, Jinan City, China) CO., Ltd.

### 2.2. Animals

Eight-week-old male BL/6 mice were obtained through and housed in cages at the Biomedical Research Association of the Academy of Athens, with controlled temperature of 22 °C and a 12-h light–dark cycle. Mice were fed standard pellets and had access to water ad libitum.

### 2.3. Experimental Design

Ethical approval was obtained from the Protocol Assessment Committee of the Department for Farming and Veterinary Services of the Decentralised Administration of Attica before commencement of the study (approval code: 453257, approval date: 7 August 2019). 55 eight-week-old male BL/6 mice were randomly assigned to 3 groups. In Group A (DSS + Que; *n* = 21), mice received Dextran Sodium Sulphate (DSS) 5% ad libitum in their drinking water plus twice daily administration of Que-HP-β-CD (Que) through gastrogavage, after reconstitution with water for injection (WFI), at a dosage of 0.5 mg/kg (administered volume of 0.25 mL). In Group B (DSS; *n* = 20), mice received DSS 5% ad libitum in their drinking water plus twice daily administration of 0.25 mL HP-β-CD aqueous solution through gastrogavage. In Group C (control; *n* = 14), mice had free access to drinking water and no other intervention. All animals received standard food pellets.

Administration of 5% DSS in the drinking water of mice is a well-established model of experimental colitis first proposed by Okayashu in 1985 [44] and widely used since then [45]. DSS disrupts the intestinal lining and allows the translocation of gut microbiota and other proinflammatory factors through the epithelial barrier, thus promoting inflammation [46]. Animals treated with DSS develop severe symptoms of colitis within a week, along with macrophage activation, colonic inflammation and loss of epithelial integrity.

All mice were weighed daily. Evaluation for colitis was done by examination of stool consistency and the presence of faecal blood was noted for each cage of animals. These were used to calculate the Disease Activity Index (DAI) for each mouse, using the established scale [47] outlined below.

On day seven, all mice were anesthetised, a blood sample was obtained by cardiac puncture and the mice were euthanised using cervical dislocation. The large intestine was excised from the ileocecal junction to the rectum, measured, and samples were gathered for histological and immunochemical analysis.

### 2.4. Disease Activity Index

For evaluation of the disease activity, an established Disease Activity Index (DAI) was used [47]. Weight decrease, stool consistency and presence of blood in the stool were scored separately (Table 1). The individual scores were combined and then divided by three, with the subsequent number being the DAI score for each mouse. DAI values were calculated on a daily basis, starting on day two.

The clinical parameters used in the DAI are functional measures that are somewhat analogous to clinical symptoms in human IBD, It is reported by Murphy et al. that this method of scoring has minimum variations (coefficient of variation = 4.8%, *n* = 144) and that the scores correlated well with more specific scores of inflammation, such as crypt scores [48]. 

### 2.5. Tissue Collection and Histology

On day seven, animals were anesthetised, a blood sample was taken by cardiac puncture and the animals were euthanised using cervical dislocation. Immediately after euthanasia, the abdominal cavity was opened, and the large intestine was excised from the ileocecal junction to the rectum. The length of the bowel was measured, as a shorter bowel length is observed in inflammatory conditions.

Samples of the ascending, transverse, descending colon as well as from the rectum were cut and either fixed in 10% formalin buffered phosphate-buffered saline (PBS) or flash-frozen in liquid nitrogen for immunological investigation.

The formalin-fixed samples were cut and stained with hematoxylin and eosin. Sections were graded for inflammatory activity as inactive (absence of neutrophils), mild (activity involving <50% of the mucosa), moderate (activity involving >50% of the mucosa; crypt abscesses usually seen at this grade) and severe (presence of surface ulceration or erosion). A Nikon Eclipse 50i microscope (Nikon Instruments Co., Ltd., Shanghai, China) was used, with magnification/aperture 20×/0.40 and 40×/0.65. Pylon Viewer Version 5.0.12.11830 64 Bit (Basler Inc., Ahrensburg, Germany) was used for the acquisition of images.

#### 2.5.1. Immunohistochemistry for Integrin a4β7

Sections, 2μm thick, were deparaffinized and placed in target-retrieval-solution (ph9) (DAKO, Santa Clara, CA, USA) followed by microwaving (3 × 5 min). This was followed by incubation with the primary antibody (Integrin alpha4 + beta7 ab73261 Abcam Rat Monoclonal/RM0059–2G18/LOT:GR3329382-2) at a dilution of 1:100 overnight at 4 °C. Sections were then washed with PBS and blocked with peroxidase (peroxidase blocking reagent SM801, DAKO, Santa Clara, United States) for 10 min. Sections were then incubated for 30 min with Rabbit anti-Rat IgG(H-L) Secondary (Biotin) NBP1-75421(NOVUS, Bio-Techne Ltd., Oxon, UK). They were then washed again with PBS and the secondary antibody (EnVisionFlex/HRPSM802, DAKO, Santa Clara, CA, USA) was added for 30 min. The slides were then coloured with 3,3-diaminobenzidine (DAB) and sections were counter-stained with hematoxylin. A Nikon Eclipse 50i microscope (Nikon Instruments Co., Ltd., Shanghai, China) was used, with magnification/aperture 20×/0.40 and 40×/0.65. Pylon Viewer Version 5.0.12.11830 64 Bit (Basler Inc., Ahrensburg, Germany) was used for the acquisition of images

#### 2.5.2. Immunohistochemistry for MAdCAM and CD11b

Sections, 2 μm thick, were deparaffinized and placed in target-retrieval-solution (ph9) (DAKO, Santa Clara, CA, USA) followed by microwaving (3 × 5 min). This was followed by incubation with the primary antibody (MAdCAM1 ab214198 Abcam Rabbit Polyclonal/LOT:GR3268444-3, CD11b ab224800 Abcam Rabbit Monoclonal/SP 331 LOT: GR3221248-3) at a dilution of 1:100 overnight at 4°C. Sections were then washed with PBS and blocked with peroxidase (peroxidase blocking reagent SM801, DAKO) for 10 min. They were then washed again with PBS and the secondary antibody (EnVisionFlex/HRPSM802, DAKO) was added for 30 min. The slides were then coloured with 3,3-diaminobenzidine (DAB) and sections were counter-stained with hematoxylin. A Nikon Eclipse 50i microscope (Nikon Instruments Co., Ltd., Shanghai, China) was used, with magnification/aperture 20×/0.40 and 40×/0.65. Pylon Viewer Version 5.0.12.11830 64 Bit (Basler Inc., Ahrensburg, Germany) was used for the acquisition of images.

### 2.6. Statistical Analysis

Continuous variables are summarized using appropriate location (e.g., mean) and dispersion (e.g., standard deviation (SD)) measures. Categorical data are presented as frequencies and percentages. Missing data are excluded from all analyses. For longitudinal continuous outcomes the Generalized Least Square (GLS) approach was used for time trend comparison across groups allowing for changes in variability over time. For ordinal categorical responses, the proportional odds approach was employed to evaluate the association of the response with covariates. Non-linear relationships were also explored.

Repeated measurements are available for weight and DAI over a 7-day period. Initially, pairwise comparisons among the three groups for each day were performed for the average weight and DAI readings, using a t-test. A similar approach was employed for comparing the standard deviation across the three groups at each time point on an appropriate scale. Time evolution of the mean and the standard deviation of weight and DAI was initially assessed separately. These results were used as input to a GLS model, where the time course of the average and the SD of weight/DAI were modeled simultaneously. The final time-trend of the mean and the SD for weight/DAI for each group were quantified as an output of the GLS approach.

For bowel length, histology, a4b7, CD11b and MADCAM, univariate pairwise comparisons across the three groups were initially performed. For continuous post-euthanasia outcomes (a4b7, CD11b and MADCAM), appropriate regression methods were employed to reveal the association of the response with DAI and group membership. A proportional odds model was employed to explore the association of group membership with the level of the histology score. Due to the small sample size (*n* = 28), the significance level was set at 10% on all statistical tests performed for this analysis.

All analyses were performed in R (version 4.1.3) (R Foundation, Vienna, Austria). The level of statistical significance was set at 5%, unless stated otherwise.

## 3. Results

Weight was measured daily for all 55 animals (Group A (DSS + Que): *n* = 21; Group B (DSS): *n* = 20; Group C (Control): *n* = 14), and DAI was calculated for each animal. Bowel length was measured for 39 animals (Group A: *n* = 17; B: *n* = 16; C: *n* = 6), histological score was calculated for 29 animals (A: *n* = 12; B: *n* = 12; C: *n* = 5) and immunological analysis was conducted on 30 (A: *n* = 12; B: *n* = 12; C: *n* = 6).

### 3.1. Weight

Pairwise comparisons for the average weight within any day, showed no significant difference across the three groups.

However, when considering the percentile change of each group’s mean weight, with mean weight on day 1 being 100%, the time trend for weight loss for Group A is larger compared to the time trend for Group C by 0.47 units, whilst the time trend for Group B is even larger by 0.53 units compared to Group C (Figure 2).

### 3.2. Disease Activity Index

Pairwise univariate comparisons revealed that average DAI for Groups A (DSS + Que) and B (DSS) was significantly higher than the average DAI for Group C, after day 4 (Figure 3). Furthermore, Group B (DSS) is statistically significantly above Group A regarding average DAI during days 6 and 7.

The time course of average DAI (Figure 3) revealed that while for Group C there is no change over the time of the study, for the other two groups there is an increase in average DAI with time, with Group B reporting the sharpest rise between days 5 and 6. The evolution of DAI variation with time, a model on the logarithm of DAI SD, showed there was a common upward trend of SD with time on this scale.

This model incorporates distinct linear time trends for the average DAI for the three groups. There is a sharp increment in time for the average and the variation of DAI for Group B (DSS), while for Group A (DSS + Que) the corresponding growth is moderate and for Group C (control) the DAI mean, and variance remain almost constant in time. Considering Group C as control, the slope of the average DAI over time for Group B is greater compared with Group 3 by 0.53 DAI points per day (95% CI 0.47–0.59, *p* < 0.01), while the corresponding value for Group A is 0.26 DAI points per day compared with Group C (95% CI 0.22–0.31, *p* < 0.01).

### 3.3. Bowel Length

One day 7, the bowel length of Group A (DSS + Que) was shorter by 9 mm (95% CI 0.93–16.38, *p* = 0.04) than Group C (Control), while for Group B (DSS) the corresponding reduction in bowel length was 20 mm (95% CI 11.48–28.69, *p* < 0.01) compared with Group C. Moreover, Group B had a shorter bowel length by 11 mm (95% CI 5.45–17.40, *p* < 0.01) than Group A (Figure 4. Furthermore, higher DAI was associated with shorter bowel length: an increment in DAI index by one unit was associated with a reduction in bowel length of 16 mm (95% CI 10.88–21.91, *p* < 0.01). Variation in bowel length was statistically similar across the three groups.

### 3.4. Histology and Immunohistochemistry

Group A was 6.37 times (95% CI 0.76–53.71, *p* = 0.09) more likely to report a worse histology score than Group C, while Group B was 5.36 times (95% CI 0.73–39.69, *p* = 0.10) more likely than Group C. The histology score distributions were comparable for Groups A and B (19% more likely to finding a higher score for Group A vs. B, *p* = 0.83).

Exploration of the association of DAI index with histology score found a positive, but not statistically significant, relationship with an increase in DAI index by one unit raising the odds of a worse histology score by 2.10 times (95% CI 0.45–9.82, *p* = 0.34) (Figure 5). Using this model to predict the histology score at various DAI levels, it is evident that for small DAI (i.e., Group C) there is a higher likelihood of a lower level of histology score, for medium DAI (i.e., Group A) an equal probability at the three levels of histology score, while for larger DAI values (Group B) it is expected that a higher histology score will be found.

An initial graphical evaluation of integrin a4b7 (Figure 6a)showed a distinct difference in location and variation between the three groups (Figure 6b). The distribution of integrin a4b7 is skewed to the left; however, after a logarithmic transformation it becomes more symmetrical. On the logarithmic scale, the variances between the groups can be considered equal. A linear model (Figure 6c), with a common variance, was fitted on the logarithmic scale and the results were back transformed on the original scale. The average integrin a4b7 value for Group A is 3.71 times higher that the corresponding average for Group C (95% CI 2.19–6.27, *p*< 0.01), while the ratio of the averages between Groups B and C was estimated at 2.21 (95% CI 1.32–3.71, *p* < 0.01). When comparing Group A and B, the average for Group A was higher than that of Group B by 1.68-fold (95% CI 1.09–2.58, *p* = 0.02).

There was no significant difference in the mean and variances for MADCAM (Figure 7a) between the three groups (Figure 7b). Exploring the association between MADCAM and DAI, no linear or non-linear relationship was found.

Finally, analysis for CD11b expression (Figure 8a) shows a clear difference in location between the three groups (Figure 8b), while variances were comparable. In a typical ANOVA model, the average CD11b for Group A was the highest among the three groups, with the difference from Group B being statistically non-significant and equal to 3.56 (95% CI −2.25–9.37, *p* = 0.21), and the difference from Group C being statistically significant at 11.06 (95% CI 3.99–18.13, *p* < 0.01). Average CD11b expression for Group B was significantly higher than Group C by 7.50 (95% CI 0.54–14.46, *p* = 0.04). Relating the CD11b measurements to DAI scores, the association was described by a quadratic concave curve with a maximum of CD11b = 23.1, when DAI = 0.7.

## 4. Discussion

In the present study we employed an experimental model of colitis in mice to investigate the potential protective antioxidant properties of quercetin. Quercetin is a flavonoid that has been shown to have protective effects against inflammatory conditions of various diseases, such as asthma [49], atherosclerosis [50], arthritis [51] and dermatitis [52]. The antioxidant properties of quercetin are well established [29], making it a viable candidate for investigation in conditions where oxidative stress is implicated in disease pathogenesis and related tissue injury.

Previous studies have indicated an imbalance in the oxidative status of the bowel of patients with inflammatory bowel disease [53]. Reactive oxygen species, molecules that damage DNA and proteins and have been implicated in the pathogenesis [3] and in contributing to tissue damage in UC, are produced in greater quantities in the colonic mucosa of patients with UC compared with healthy individuals. UC is characterized by an up-regulation of the synthesis and release of a variety of pro-inflammatory mediators, such as eicosanoids, platelet activating factor, reactive oxygen and nitrogen metabolites and cytokines, thus influencing mucosal integrity and leading to excessive injury. The cell types involved in the mucosal inflammatory response are similar to those found at systemic inflammatory sites including macrophages. Comolada et al. demonstrated that the anti-inflammatory effects of quercetin in Inflammatory Bowel Disease (IBD) is mediated through inhibition of the NF-κB pathway, which pathway has a role in the expression of proinflammatory genes [54].

In our study, animals treated with DSS developed symptoms of colitis, with an increase in their DAI score. More specifically, both Groups A and B started increasing their DAI score from day three, with Group B, as expected, showing a notable increase in DAI score on day five and this tripled in value by day seven. In Group A, those receiving DSS and quercetin, the DAI score increased at a slower rate than Group B. Animals in Group C, the control group, retained a very low DAI score, as expected. These results indicate a protective effect of quercetin on treated animals, as their DAI scores increased more slowly than those receiving DSS but no quercetin.

The average weight reduction over time did not statistically differ between the three Groups. However, the time trend for weight reduction in Group B was numerically larger than Group A and even more so than the control Group C, allowing for a hypothesis of a type II statistical error, i.e., a larger number of animals would have allowed the attainment of statistical significance in the comparisons of interest. Importantly, the present study results indicate trends that are in the “right” direction. In addition, a longer duration, or a higher dose of DSS, or both, might have been beneficial in providing more time for the disease to develop and allowing for the potential further differentiation of results of the three treatment groups. In addition, as both Groups A and B lost weight at a similar rate but the DAI score of Group B is markedly higher than that of Group A, it can be concluded that the animals in Group B developed more severe symptoms of diarrhea and blood in the stools and that a more severe loss of weight might follow should the study have been performed over a longer period of time.

Bowel length was significantly shorter in the DSS groups than the control group, with the quercetin-treated group showing intermediate results. This is indicative of greater inflammation in the bowel of the animals of Group B, whereas the results of Group A suggest that the course of the disease was slowed by the administration of quercetin.

The promising disease markers were not reflected in the findings of the histological analysis, where no difference was observed between Groups A and B. However, the fact that compounds with similar properties have been proven therapeutic for UC [17] and the promising result of the clinical evaluation leads us to believe that the histological results are simply because of the small sample size involved. Indeed, the model combining histological distribution and expected DAI yielded results that were more aligned with the potential antioxidant effect of quercetin, with an increase in DAI, a trait more representative of the animals of Group B, predicting a higher histological score.

In terms of immunohistochemistry, CD11b is a subunit that together with CD18 forms the integrin Mac-1. It is present in macrophages, dendritic cells and in B cells of the peritoneal cavity. Studies have shown that CD11b ameliorates symptoms of colitis through IL-10 production [55]; therefore, in mice from Group A (DSS + Que), which showed greater values of CD11b expression, there would appear to be better mobilization of the immune system, which would attenuate the symptoms of the disease.

Integrin a4/b7 is a homing molecule which facilitates the adhesion of leukocytes to the endothelium of the blood vessels of the gut, through binding with MadCAM1, and is the target of monoclonal antibody therapies for IBD, such as vedalizumab. In our study, we found higher values of a4b7 as DAI index rose, confirming the integrin’s role in disease pathogenesis, however, the higher values found in Group A suggest that quercetin’s beneficial effects target different pathways than the integrin a4/b7-MadCAM1 axis.

Mucosal vascular addressing cell adhesion molecule-1 (MadCAM1) is specifically expressed on the endothelial cells of gut-associated lymphoid tissue and, through binding with integrin a4/β7, plays a critical role in the migration of leukocytes to the gut. It is constantly expressed at mesenteric lymph nodes and Peyer’s Patches and is upregulated during inflammation. However, there is no statistical difference in expression of MadCAM-1 between samples from patients with active disease and patients in remission [56]. Therefore, the high values found in both Groups A and B would suggest only that colitis was achieved in this model over the time period evaluated rather than levels of severity of the disease.

There were certain limitations to our study. The number of animals used was limited which, combined with the fact that it was not possible to conduct histological and immunological analysis of samples from all animals, very likely caused statistical irregularities and potentially impacted on the clarity of the results. In addition, the study would benefit from examining the effect of different doses of quercetin to determine an optimal dose for this model. As mentioned earlier, the study would also benefit from being of a longer duration, which would hopefully allow for better separation of the clinical manifestations between the groups.

Based on our results, which indicate reduced disease activity in animals treated with quercetin, this compound is a possible candidate for oral supplementation in patients suffering from UC and possibly from other inflammatory conditions of the bowel, such as Crohn’s Disease and necrotizing enterocolitis. However, further research is required.

## 5. Conclusions

The administration of quercetin, in the form of the water-soluble lyophilized complex with hydroxypropyl-β-cyclodextrin (Que-HP-β-CD), in an Ulcerative Colitis model in mice, indicates a prophylactic effect in terms of disease activity and bowel length. Our findings may find application in human disease if future investigations both in animals and humans are conducted and confirm safety and benefit of appropriate doses.

## Figures and Tables

**Figure 1 medicina-59-00087-f001:**
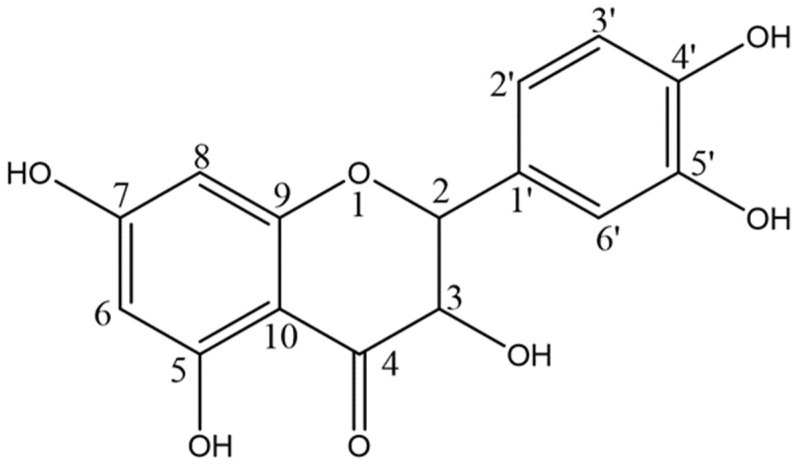
Chemical Structure of quercetin.

**Figure 2 medicina-59-00087-f002:**
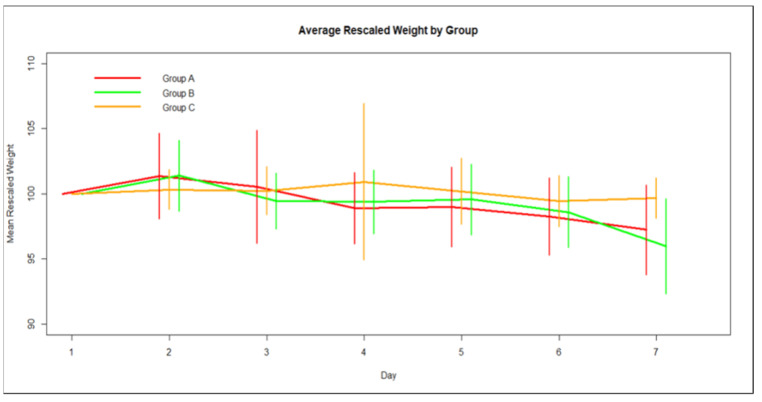
Average group percentile weight change over time and Standard Deviation.

**Figure 3 medicina-59-00087-f003:**
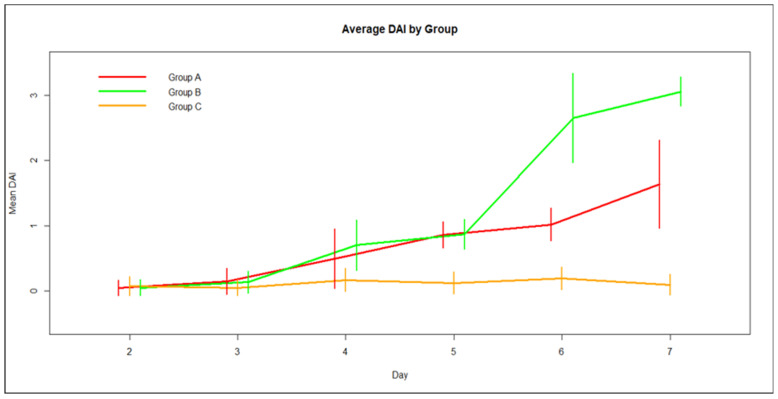
Average group DAI score over time and Standard Deviation.

**Figure 4 medicina-59-00087-f004:**
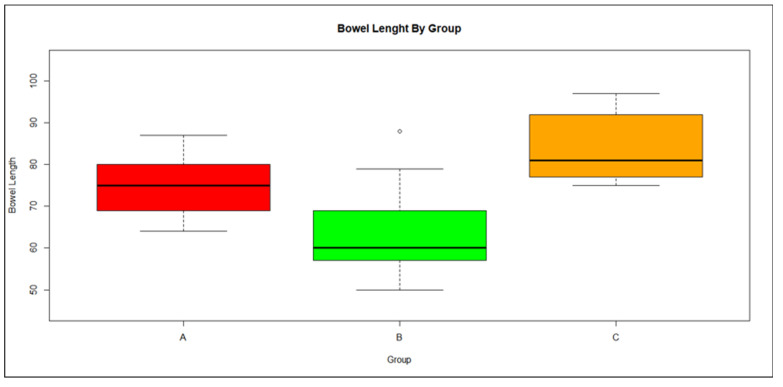
Boxplot of bowel length distribution by group showing median and range;., Red: Group A, Green: Group B, Orange: Group C.

**Figure 5 medicina-59-00087-f005:**
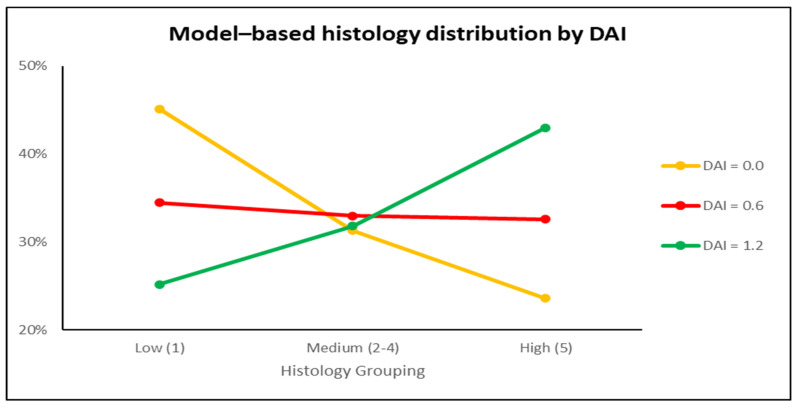
Proportional odds model of expected histology score for a specific DAI score.

**Figure 6 medicina-59-00087-f006:**
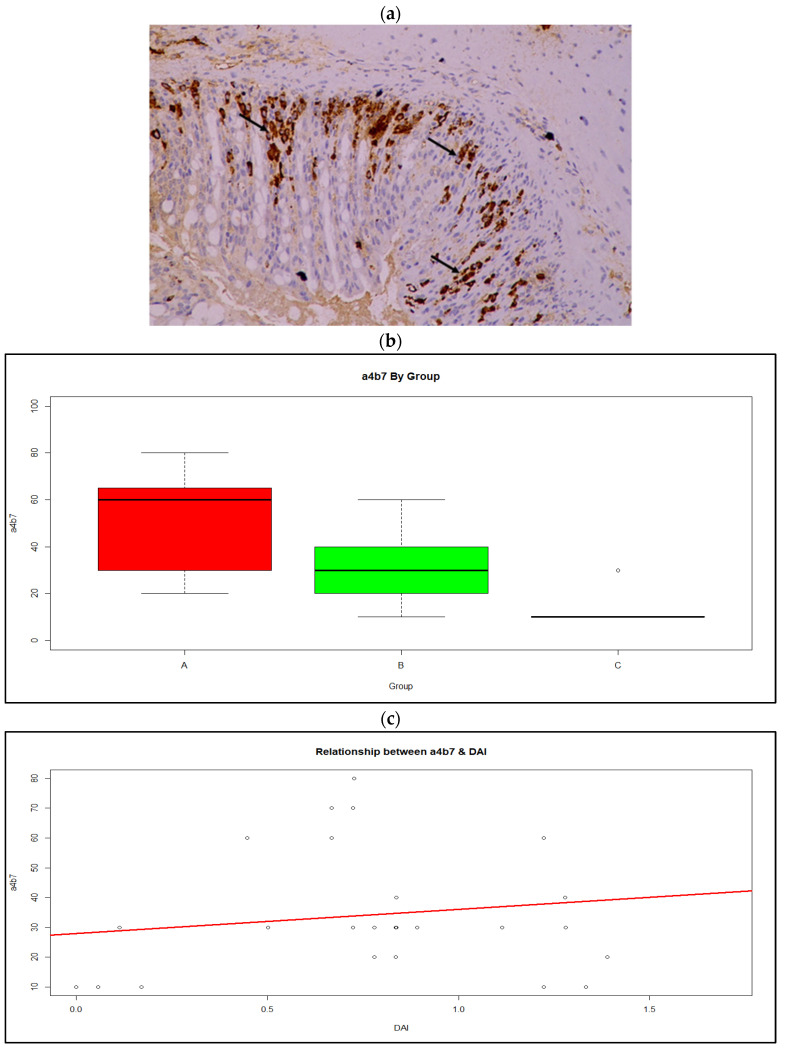
(**a**) Integrin α4β7 expression by lymphocytes of the intestinal mucosa (arrows); Magnification ×20, (b) Boxplot of a4b7 distribution by group showing median and range; Red: Group A, Green: Group B, Orange: Group C (**c**) Relationship between a4b7 expression and DAI score.

**Figure 7 medicina-59-00087-f007:**
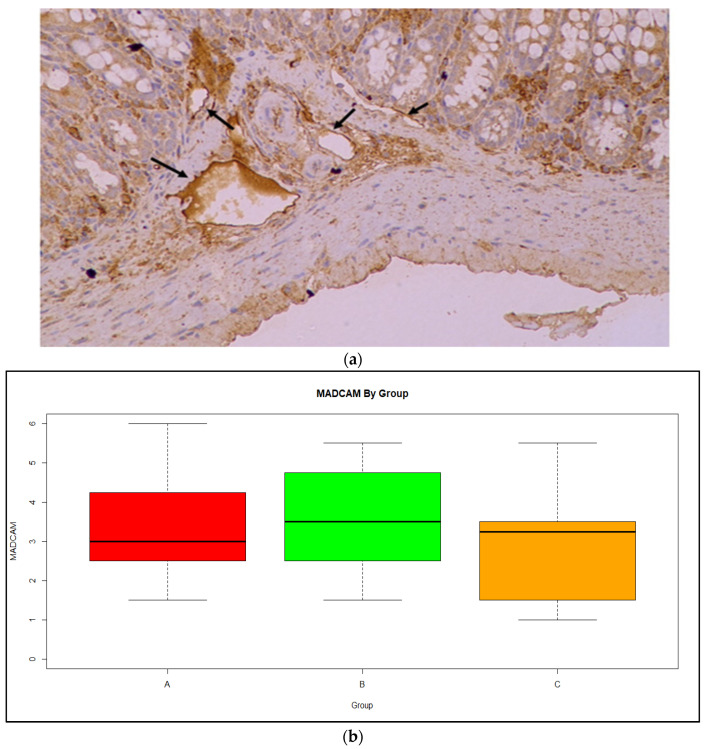
(**a**) MADCAM expression in in the vessels of the intestinal mucosa (arrows); Magnification ×20 (**b**) Boxplot of MadCAM expression by group showing median and range. Red: Group A, Green: Group B, Orange: Group C.

**Figure 8 medicina-59-00087-f008:**
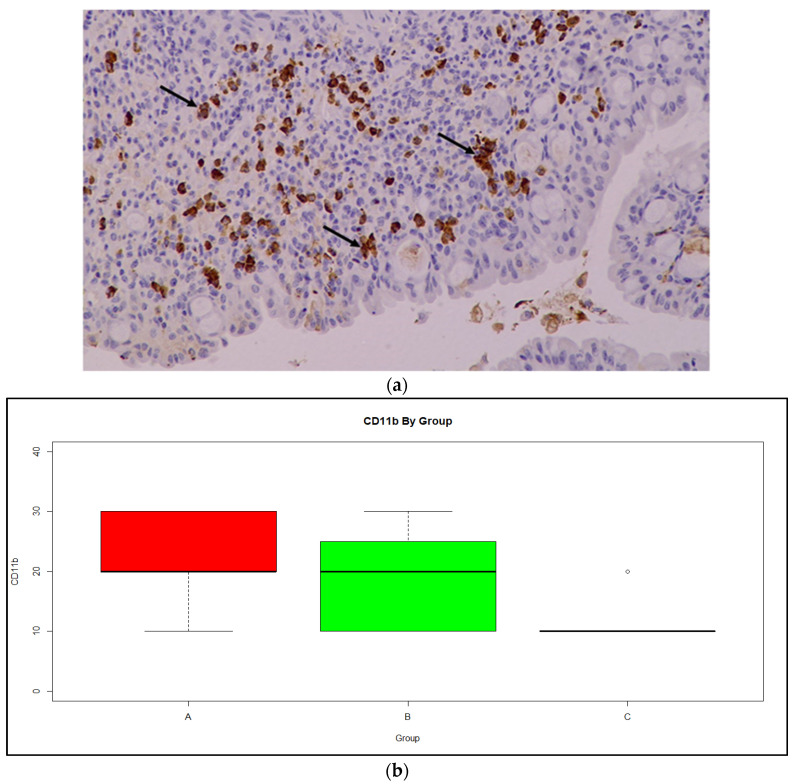
(**a**) CD11b expression in monocytes of the intestinal mucosa (arrows); Magnification ×20 (**b**) Boxplot of CD11b expression by group, showing median and range. Red: Group A, Green: Group B, Orange: Group C.

**Table 1 medicina-59-00087-t001:** Disease Activity Index [47].

Score	Body Weight Decrease (%)	Stool Consistency	Presence of Blood
0	<1	Normal	Normal
1	1–5		
2	5–10	Loose stools	
3	10–20		
4	>20	Diarrhoea	Gross bleeding

## Data Availability

The data presented in this study is available by request from the corresponding author.

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
