# Peer review of "Effects of the Antioxidant Quercetin in an Experimental Model of Ulcerative Colitis in Mice"

_medicina, 2022, doi:10.3390/medicina59010087_

Round 1

Reviewer 1 Report

In this manuscript authors reported effects of the antioxidant quercetin in dextran Sodium Sulphate experimental model of ulcerative colitis in mice.

Authors claimed that the administration of quercetin in an ulcerative colitis model in mice presents a therapeutic/prophylactic potential.

Despite the some interesting experimental observations manuscript needs substantial revision.

Methods also need to be written with more details to ensure reproducibility of the study; please add model/manufacturer information for the microscope you used.

Please high quality/resolution images for the microscopy data including original scale. 

Please explain how the histology score was calculated.

I would suggest to use additional methods to support the claim, e.g. flow cytometry and gene expression.

Reviewer 2 Report

Comment 1- In the introduction, I suggest writing a brief paragraph regarding ulcerative colitis in light of symptoms and types.

Comment 2- It would be good to include the effect of ulcerative colitis on blood count, C-reactive protein, etc.

Comment 3- Please include the chemical structure of quercetin that would be helpful to understand the mechanism.

Comment 4- The discussion is good, but the authors should include a brief introduction of molecular mechanisms linked with ulcerative colitis.

Comment 5- The significance of natural products with antioxidant potential should be included in the introduction section.

1.      https://doi.org/10.3390/pr8040468.

2.      doi: 10.2174/1871520620666200705220307.

Comment 6- I think that author could improve the introduction by mentioning the common medications for ulcerative colitis and their adverse effects.

Comment 7– Quercetin-hydroxypropyl-β-cyclodextrin (Que-HP-β-CD) inclusion complex, in its lyophilized form, has increased water solubility and chemical stability compared with quercetin, making it a viable option for oral administration.

The authors should briefly explain the mechanism or provide some good references.

 It is advised to include the role of inflammation in diabetes-induced hepatic injury.

Comment 8- Please provide criteria and strong references for the disease activity index linked with ulcerative colitis. In my view, the disease activity index can-not be calculated on the basis of stool consistency, bleeding, and body weight decrease, only.

Comment 9- It is well evident that dextran sulfate sodium (DSS) induces colon cancer in mice. This study shows that quercetin has a protective effect against ulcerative colitis.  It would be good to suggest the particular pathway of ulcerative colitis influenced by quercetin.

Comment 10- The presentation of data in the form of is very poor. The graphical presentation must be improved.

Comment 11- The figure ligands should be more informative.

Comment 12- The conclusion should be rewritten.

Round 2

Reviewer 1 Report

Comment 1

Methods also need to be written with more details to ensure reproducibility of the study; please add model/manufacturer information for the microscope you used.

Response 1

Thank you for your comment. As recommended, additions have been inserted in sections 2.4, 2.5, 2.6.1, 2.6.2 of the revised discussing in more detail the Disease Activity Index, sample collection, microscope specification and histological parameters.

Please provide information regarding the acquisition and processing of images: magnification, and numerical aperture of the objective lenses; acquisition software; and any software used for image processing subsequent to data acquisition.

Comment 2

Please add high quality/resolution images for the microscopy data including original scale.

Response 2

Thank you for your request. As per your suggestion, images have now been included as Figures 9a, 9b and 9c.

Despite authors claim images does not include scale; please provide images with original scale and higher magnification. Please provide representative images for each of treatment group.

Comment 3

Please explain how the histology score was calculated.

Response 3

The formalin-fixed samples were cut and stained with hematoxylin and eosin. Sections were graded for inflammatory activity as inactive (absence of neutrophils), mild (activity involving <50% of the mucosa), moderate (activity involving > 50% of the mucosa; crypt abscesses usually seen at this grade) and severe (presence of surface ulceration or erosion).

Please explain how did you judge inflammatory activity/presence of neutrophils based just on  hematoxylin and eosin staining. I suggest to add immunofluorescence multicolor staining for all markers from the same slide.
